# Clinical characteristics of COVID-19 in older adults. A retrospective study in long-term nursing homes in Catalonia

Uxío Meis-Pinheiro[1☉], Francesc Lopez-Segui[2☉], Sandra Walsh[3☉], Anton Ussi[4‡], Sebastia Santaeugenia [5,6‡], Jose Augusto Garcia-Navarro[7‡], Antonio San-Jose[8‡], Antoni L. Andreu[1,4‡], Magda Campins[9‡], Benito Almirante[10‡]*

1 Associació Catalana de Recursos Assistencials, ACRA, Barcelona, Spain, 2 Centre de Recerca en Economia de la Salut—CRES UPF, Barcelona, Spain, 3 Institut de Biologia Evolutiva (UPF-CSIC), Universitat Pompeu Fabra, Barcelona, Spain, 4 European Infrastructure for Translational Medicine, EATRIS, Amsterdam, Netherlands, 5 Central Catalonia Chronicity Research Group (C3RG), Centre for Health and Social Care Research (CESS), Universitat de Vic–University of Vic-Central University of Catalonia (UVIC-UCC), Vic, Spain, 6 Chronic Care Program, Ministry of Health, Generalitat de Catalunya, Barcelona, Spain, 7 Spanish Society of Geriatrics and Gerontology, Barcelona, Spain, 8 Geriatric Unit, Hospital Universitari Vall d´Hebron, Barcelona, Spain, 9 Preventive Medicine and Epidemiology Department, Hospital Universitari Vall d´Hebron, Barcelona, Spain, 10 Infectious Diseases Department, Hospital Universitari Vall d´Hebron, Barcelona, Spain

☉ These authors contributed equally to this work.
‡ These authors also contributed equally to this work.
* balmirante@vhebron.net

**Data Availability Statement:** All relevant data are within the paper and its S1 File.

**Funding:** The authors(s) received to specific funding for this work.

## Abstract

The natural history of COVID-19 and predictors of mortality in older adults need to be investigated to inform clinical operations and healthcare policy planning. A retrospective study took place in 80 long-term nursing homes in Catalonia, Spain collecting data from March 1st to May 31st, 2020. Demographic and clinical data from 2,092 RT-PCR confirmed cases of SARS-CoV-2 infection were registered, including structural characteristics of the facilities. Descriptive statistics to describe the demographic, clinical, and molecular characteristics of our sample were prepared, both overall and by their symptomatology was performed and an analysis of statistically significant bivariate differences and constructions of a logistic regression model were carried out to assess the relationship between variables. The incidence of the infection was 28%. 71% of the residents showed symptoms. Five major symptoms included: fever, dyspnea, dry cough, asthenia and diarrhea. Fever and dyspnea were by far the most frequent (50% and 28%, respectively). The presentation was predominantly acute and symptomatology persisted from days to weeks (mean 9.1 days, SD = 10,9). 16% of residents had confirmed pneumonia and 22% required hospitalization. The accumulated mortality rate was 21.75% (86% concentrated during the first 28 days at onset). A multivariate logistic regression analysis showed a positive predictive value for mortality for some variables such as age, pneumonia, fever, dyspnea, stupor refusal to oral intake and dementia (p<0.01 for all variables). Results suggest that density in the nursing homes did not account for differences in the incidence of the infection within the facilities. This study provides insights into the natural history of the disease in older adults with high dependency living in long-term nursing homes during the first pandemic wave of March-May 2020 in the region of

**Competing interests:** The authors have declared that no competing interests exist.

Catalonia, and suggests that some comorbidities and symptoms have a strong predictive value for mortality.

## Introduction

COVID-19 presents with a broad spectrum of severity ranging from a completely asymptomatic form to a severe acute respiratory syndrome [1]. The majority of patients presenting with COVID-19 experience a mild illness that can usually be managed in the community but in some patients clinical deterioration occurs and age and the presence of comorbidities are associated with a more severe disease and poor outcome [2]. Major clinical features of the disease include fever, dry cough and dyspnea that can lead to a severe respiratory distress in some patients, but also other signs and symptoms also occur. These include muscle or body aches, anosmia, dysgeusia, headache, gastrointestinal symptoms such as diarrhea and a wide range of skin lesions such as erythematous rashes, urticaria, and chicken pox-like vesicles [2–7]. Although early reports indicated that the main mechanism of transmission was respiratory through respiratory droplets exhaled by an infected person, current knowledge supports the theory that infection is spread through exposure to smaller virus-containing respiratory droplets and particles that can remain suspended in the air over long distances and time, a mechanism known as airborne transmission [8]. Soon after the disease appeared in Wuhan at the end of 2019, the global outbreak of the SARS-CoV2 virus created a public health emergency on a scale unprecedented in recent history. The region of Catalonia in north-eastern Spain has been severely impacted by the pandemic and, as of April 23rd, 2021, remains in the throes of the crisis with more than 590,000 infections. Of note is the high mortality rate notified by the regional health care authorities in long term care homes [9]. More than 8,798 people have died in these facilities, according to official estimates, representing 50% of the total casualties in the region, a percentage similar to other regions in Spain [10, 11]. However, in most cases, COVID-19 was not confirmed during the early stages of the outbreak in March and April 2020, as RT-PCR based diagnostic tests were not fully available. The clinical courses of some casualties in these facilities strongly suggest the presentation of COVID-19, but could not be laboratory confirmed.

The vulnerability of the elderly in long-term care facilities to respiratory disease outbreaks, including influenza and other commonly circulating human corona viruses such as the common cold, is well recognized [12, 13]. These institutions represent a risky setting for COVID-19 transmission due to the characteristics that define the setting i.e. residents who are predominantly at advanced ages and have underlying medical conditions. Although the specific elements of the viral versus host factors that define susceptibility to severe disease are not well understood [14, 15] some specific characteristics of older individuals living in nursing homes such as the high proportion of patients receiving chronic treatment with Anriotensin II receptors blockers increase the risk of acquiring SARS-CoV2 infection [16]. Also the lack of PPEs at the beginning of the pandemic outbreak and the difficulties of wearing masks in aged individuals suffering from cognitive impairment may have accounted for the fast expansion of the virus within these facilities [17]. The high incidence of COVID-19 in long-term care facilities for the elderly has generated a great deal of clinical data that has been recorded in the official data registries of these institutions. Here we report clinical and environmental retrospective data from a multicenter cohort of positive RT-PCR residents living in nursing homes of the region.

## Materials and methods

### Study population

The study was conducted in long-term nursing homes belonging to ACRA (Associació Catalana de Recursos Assistencials), the largest network of long-term care facilities in the region. These facilities provide full long-term care for older adults that have a high or very high degree of dependency. An open call to participate in the register among the 400 centers of the organization was launched on June 1st 2020, with 160 agreeing to participate. Among them, 80 had confirmed cases of COVID-19 among residents. The sample included facilities located in urban and suburban areas; 2.5% in towns up to 2,000 inhabitants; 20% between 2,001 and 10,000; 28.75% between 10,001 and 100,000 and 48.75% in towns over 100,000. In these facilities, a total of 2,092 COVID-19 cases were confirmed by a RT-PCR test. Additional residents with suspected COVID-19 identified on the basis of serological tests—but without RT-PCR test results—were obtained; however, they were not included in the study due to the high variability of the tests, poor analytical performance and lack of approval by the National Medicinal Products Agency.

### Study method/assessment

A detailed questionnaire was distributed to participating centers where information on the characteristics of the facility and pseudonymized clinical information (including when the positive RT-PCR were performed), provided retrospective data over a period of three months, from March 1st to May 31st, the period of the first pandemic wave in Spain. Clinical information was extracted from the Clinical History forms of each resident and reassessed by the medical and nursing services of the nursing home during the period in which data was collected, to define the characteristics of the natural history of the disease in the population studied. The information on the structural characteristics of the nursing-homes included: total number of places, number of beds, single and double rooms, number of bathrooms ensuite or in common areas, as well as the number and size of common areas, including the number of living units cases where in case the nursing-home was organized on the basis of this model [18].

Instead of arbitrary "large" or "small" facilities and, with the aim of differentiating facilities where all residents shared the same spaces from those where daily life is organized in subgroups of residents sharing the same common spaces, we created a synthetic indicator based on the number of "spaces" and "units". "Spaces" were defined as rooms (excluding bedrooms) where residents spend time during the day (sitting areas, TV rooms, activity rooms, dining areas and so on). "Units" were defined as living units [18], the interconnected group of spaces where independent subgroups of residents do all their daily activities, including bedrooms and common spaces used by a particular subgroup.

Based on these definitions, a synthetic indicator was defined to describe the structural characteristics of the facilities by using 7 variables:

a. No. of spaces between 10 and 19 m2

b. No. of spaces between 20 and 49 m2

c. No. of spaces between 50 and 100 m2

d. No. of spaces of more than 100 m2

e. No. units of less than 10 people

f. No. of units between 10 and 15 people

g. No. of units of more than 15 people

The first four variables, which refer to the surfaces were transformed by dividing them by the total number of spaces (No. of spaces between. . ./No. of total spaces). Similarly, the three variables related to the number of people per unit are transformed by dividing them by the total number of units (No. of units between. . ./No. of total units). This transformation provides and indicator of the proportion of spaces and units of each type. Variables are weighted with values ranging from 1 to 4, being 1 "low density" and 4 "high density" resulting in a coefficient indicative of density per living unit in the entire facility.

The workforce/resident ratio was considered a non-significant variable as it was consistently maintained among these facilities following mandatory regional social care legislation.

Anonymized demographic and clinical data from all positive RT-PCR residents were collected from the institutional records. Demographic variables included age and sex. Clinical variables included the presence or absence of signs and symptoms that have been reported to be associated to COVID-19 [19–22], such as: fever, dyspnea, asthenia, cough, muscular pain, rhinorrhea, sore throat, diarrhea, vomiting, refusal to intake, skin vesicles or eczemas, insomnia, confusion and stupor. Other symptoms associated with COVID-19 such as anosmia or dysgeusia were not recorded as the frailty and high level of dependency of the population studied, including variable levels of cognitive impairment and dementia would had introduced significant bias in the interpretation. The presence or absence of Rx- confirmed pneumonia was also recorded. For each symptom the date of onset and the date of finalization was confirmed. Clinical data also include the presence of frequent chronic underlying health conditions. Clinical follow-up included information on hospitalization and also the resolution of the disease: recovery, ongoing clinical course or exitus at the time data were collected. The date when a positive RT-PCR test was obtained and, in most cases, when it turned to negative was also collected.

Date at onset was defined for each resident as the day the first symptom was present, if this symptom was present within a period of 14 days prior to or after a positive RT-PCR test.

## Statistical analysis

We computed descriptive statistics to characterize the demographics and clinical variables of our sample overall and by their symptomatology. Statistically significant bivariate differences were assessed using t-test, Wilcoxon rank sum test or Fisher's exact test. Additionally, we constructed a logistic regression model to assess the relationship between exitus and a number of variables, including age, symptomatology and underlying health conditions. Microsoft Excel and R-4.0.2. were used for data processing and quantitative analysis.

## Ethical issues

The study was approved by the Research Ethics Board of the Hospital Vall d´Hebron in Barcelona, Catalonia, Spain, a reference healthcare institution in the region for COVID-19 patients. A waiver of informed consent was granted because the data were collected for public health surveillance purposes.

## Results

The cohort described here included 2,092 patients, residents of long-term nursing homes that were infected by SARS-CoV2 during the first wave of the pandemic in Catalonia (March-April 2020). Mean age of the cohort was 86.7 (SD, 7.06) with a higher proportion of females (73% of total), consistent with the demographic characteristics of this population in nursing homes in Catalonia [9]. Overall, the demographic characteristics of residents and the facilities faithfully

represent the general attributes of nursing-homes within the region, and the sample accounts for approximately 10% of the total number of long-term care facilities in Catalonia [23].

Table 1 shows demographic data and clinical symptoms of RT-PCR positive residents. The average incidence of positive RT-PCR in the total population of 80 facilities was 28% (ranging from 1% to 71%) of the total number of residents, and no regional significant differences were identified (data not shown).

**Table 1. Demographics, clinical features and underlined health conditions of the cohort.**

|  | Total N = 2092 | Exitus N = 455 | Survivor N = 1637 | P-value |
|---|---|---|---|---|
| **Demographic variables** | | | | |
| Age (SD) | 86.7 (7.06) | 87.97 (6.65) | 86.3 (7.82) | 0.0002 |
| Gender (Female) | 1532 (73.23%) | 287 (63.08%) | 1245 (76.05%) | <0.0001 |
| Complex and chronic patient | 792 (38%) | 158 (34,7%) | 634 (38.73%) | 0.1262 |
| Pneumonia | 331 (16%) | 131 (28.8%) | 200 (12,22%) | <0.0001 |
| Hospitalization | 461 (22.04%) | 148 (32.5%) | 313 (19,12%) | <0.0001 |
| **Symptomatology** | | | | |
| Fever | 1055 (50.43%) | 344 (75.6%) | 711 (43.43%) | <0.0001 |
| Dyspnea | 601 (28.73%) | 268 (58.9%) | 333 (20.34%) | <0.0001 |
| Asthenia | 394 (18.83%) | 147 (32.31%) | 247 (15.09%) | <0.0001 |
| Dry cough | 399 (19.07%) | 89 (19.56%) | 310 (18.94%) | 0.787 |
| Muscle pains | 181 (8.65%) | 54 (11.87%) | 127 (7.76%) | 0.00808 |
| Nasal congestion | 71 (3.39%) | 21 (4.62%) | 50 (3.05%) | 0.108 |
| Excessive nasal discharge | 66 (3.15%) | 28 (6.15%) | 38 (2.32%) | 0.000116 |
| Sore throat | 47 (2.25%) | 11 (2.42%) | 36 (2.2%) | 0.724 |
| Diarrhea | 330 (15.77%) | 66 (14.51%) | 264 (16.13%) | 0.425 |
| Vomiting | 184 (8.8%) | 47 (10.33%) | 137 (8.37%) | 0.191 |
| Refusal to oral intake | 314 (15.01%) | 165 (36.26%) | 149 (9.1%) | <0.0001 |
| Lip blisters | 3 (0.14%) | 2 (0.44%) | 1 (0.06%) | 0.121 |
| Confusion | 166 (7.93%) | 55 (12.09%) | 111 (6.78%) | 0.000387 |
| Stupor | 121 (5.78%) | 103 (22.64%) | 18 (1.1%) | <0.0001 |
| Insomnia | 25 (1.2%) | 8 (1.76%) | 17 (1.04%) | 0.223 |
| Eczema | 40 (1.91%) | 4 (0.88%) | 36 (2.2%) | 0.0806 |
| **Underlying health conditions** | | | | |
| Dementia | 1243 (59,41) | 326 (71,65%) | 917 (56,02%) | <0.0001 |
| Heart disease | 1007 (48.14%) | 204 (44.84%) | 803 (49.05%) | 0.112 |
| Cerebrovascular disease | 483 (23.09%) | 117 (25.71%) | 366 (22.36%) | 0.148 |
| DM without organic involvement | 400 (19.12%) | 96 (21.1%) | 304 (18.57%) | 0.226 |
| DM with organic involvement | 168 (8.03%) | 35 (7.69%) | 133 (8.12%) | 0.845 |
| COPD | 275 (13.15%) | 67 (14.73%) | 208 (12.71%) | 0.272 |
| Hepatopathy | 90 (4.3%) | 13 (2.86%) | 77 (4.7%) | 0.0904 |
| Peptic ulcer | 89 (4.25%) | 27 (5.93%) | 62 (3.79%) | 0.0492 |
| CKD | 501 (23.95%) | 114 (25.05%) | 387 (23.64%) | 0.535 |
| Connective tissue disease | 115 (5.5%) | 24 (5.27%) | 91 (5.56%) | 0.908 |
| Tumor without metastases | 228 (10.9%) | 41 (9.01%) | 187 (11.42%) | 0.149 |
| Solid Tumor with metastases | 14 (0.67%) | 4 (0.88%) | 10 (0.61%) | 0.52 |
| Hematological tumor | 16 (0.76%) | 1 (0.22%) | 15 (0.92%) | 0.22 |
| AIDS | 5 (0.24%) | 0 (0%) | 5 (0.31%) | 0.592 |

Legend: N(%). P-values were calculated by Wilcoxon Rank Sum test and Fisher's exact test.

DM = Diabetes Mellitus; COPD = Chronic Obstructive Pulmonary Disease; CKD = Chronic Kidney Disease, AIDS = Acquired Immune Deficiency Syndrome.

Fig 1 shows the clinical characteristics of the cohort. 71% of residents showed symptoms and/or signs associated with COVID-19 and 29% were completely asymptomatic. Fever and dyspnea were by far the most frequent symptoms (in 50% and 28% of the cohort, see Fig 1A). When we analyzed the frequency of symptoms in patients that were symptomatic (excluding the 29% of patients that were completely asymptomatic) the frequency of fever and dyspnea was even higher (71% and 40%). Other relatively frequent symptoms or signs included persistent cough, asthenia and diarrhea. The proportion of individuals with three of more symptoms compatible with COVID-19 was 31%. Fig 1B also shows the incidence of clinical phenotypes resulting from the combination of the five most frequent symptoms (fever, dyspnea, dry cough, asthenia and diarrhea). 67.78% patients showed clinical presentations resulting from different combinations of these five major symptoms. Strong correlations for the pairs fever-dyspnea (CC 0.33, p- < 0.001), fever-asthenia (CC 0.22, p < 0.001), dyspnea-asthenia (CC 0.24, p < 0.001) and asthenia-refusal to oral intake (CC 0.39, p-value p < 0.001) were observed.

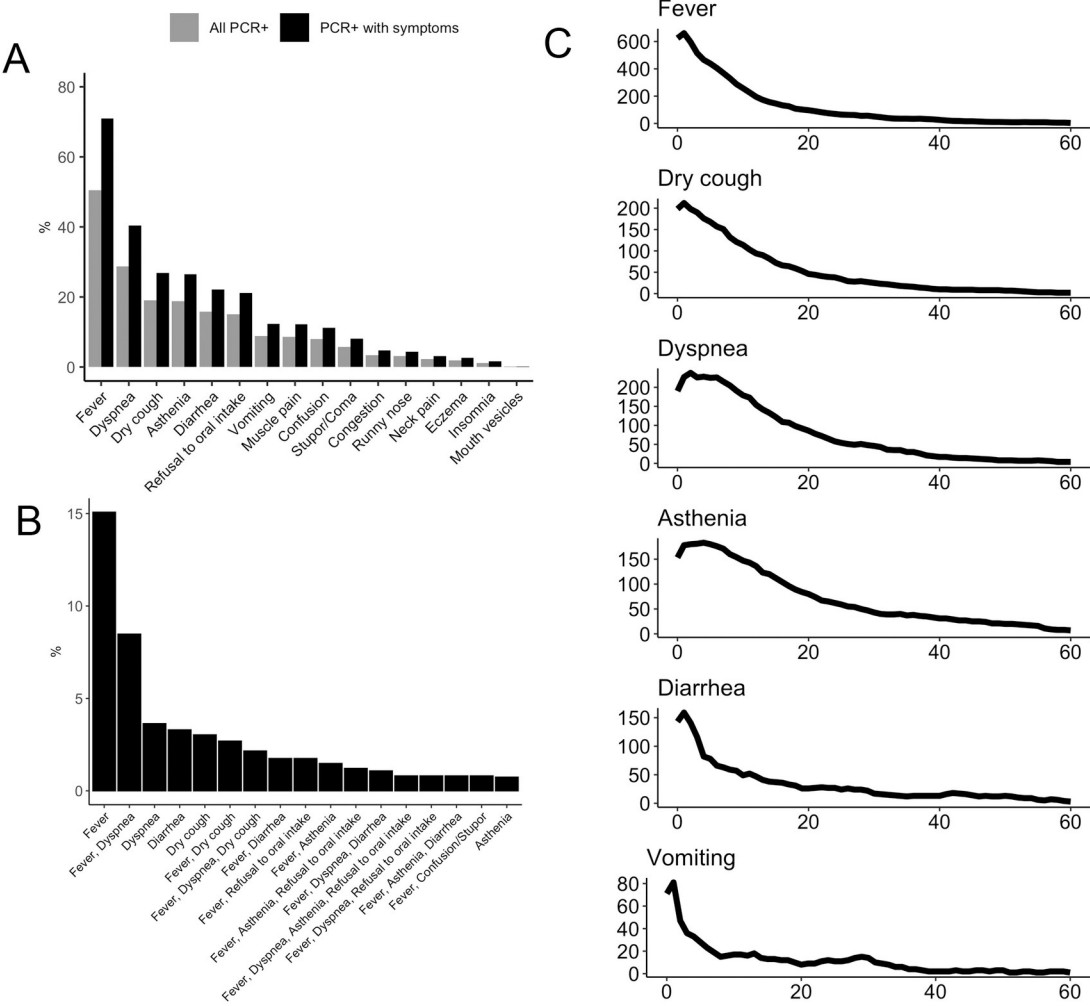

**Fig 1. Clinical features of the cohort.** Legend: Panel A- Percentage of patients presenting a symptom. Lights bars indicate percentage of symptoms in all positive PCR individuals. Dark bars indicate percentage of symptoms in positive PCR individuals who show symptoms. Panel B- Most frequent clinical phenotypes among PCR positive patients. Percentage of the most frequent clinical phenotypes in PCR positive patients with symptoms. Panel C- Clinical courses of six of the most frequent symptoms from the onset of the disease. Duration (in days) of the most frequent symptoms (Date at onset is defined for each resident as the day the first symptom was present).

A significant correlation was also identified for the pair refusal to oral intake and stupor (CC 0.33, p < 0.001), in particular in patients with a poor clinical outcome.

A longitudinal follow-up of the symptoms from onset showed a predominantly acute presentation of the symptomatology which persisted for a few days to a few weeks (see Fig 1C). Fever, diarrhea and vomiting lasted in most cases between 1 and 2 weeks (8, 5.7 and 3.7 mean days respectively), while dyspnea and asthenia tended to last longer (9.6 and 14.3 mean days). 16% of residents had confirmed pneumonia and 22% required hospitalization.

The accumulated mortality rate in this cohort at three months was 22% and there were significant differences between patients who were hospitalized and those who were not (32% vs. 18%, p<0.01). Fig 2A shows the cumulative mortality, which accounted for up to 100% during the 80 days after onset, in particular during the first 28 days, which accounted for 86% of the casualties. The massive and acute outbreak of the disease in the region resulted in a concentration of casualties (see Fig 2B) during a period of 2 months (from mid-March to mid-May), the majority (77%) concentrated in a period of 1 month from end of March to the end of April.

Table 2 presents a logistic regression analysis to identify potential predictive factors for exitus. Age and the presence of pneumonia had a significant predictive value (OR = 1,03 and OR 1,74, respectively). However, the strongest correlations were identified for two symptoms: Fever (OR = 2,39) and dyspnea (OR = 3,13) together with the presence of stupor (OR = 11,38) and refusal to oral intake (OR = 3,21). Most comorbidities did not have a predictive value except dementia and hepatopathy (OR = 1,63 and OR = 0,45, respectively). Interestingly, previous conditions such as cardiovascular disease or COPD were not suggestive of increased probability of mortality.

The number of places of the facilities ranged from 15 to 300 (mean = 69,06, SD = 34,32) and did not show a significant correlation with the incidence of COVID-19 in those nursing homes that reported at least one confirmed case of the disease (Fig 3A, R = 0.098, p = 0.33). The density of the institution assessed by a synthetic indicator that weighed the number and size of living units (see methods) showed also no correlation with the incidence of the disease (Fig 3B, R = 0.83, p = 0.47), suggesting that nursing homes organized on a model based on living units were not prone to have a higher incidence of the disease when compared to nursing homes where common spaces where shared by the entire population, once the virus had been introduced in the facility.

## Discussion

COVID-19 exacted a heavy toll in nursing home facilities in Spain, causing a high number of deaths [10]. Restriction policies and lockdowns were not imposed until the Government declared a state of emergency on March 14, 2020 [24]. In addition to the age and comorbidity profiles of residents, other variables make these institutions particularly fragile when dealing with an infectious disease outbreak. These include lack of access to testing and personal protective equipment, the close quarters of residents, the difficulty of maintaining social distance among mobile patients with cognitive impairment and a workforce that has not been extensively trained for managing infections [25–27]. This combination of factors (intensity of the pandemic outbreak, lack of molecular diagnostic tools and protective equipment, and lack of training of the workforce) created a "perfect storm" that may explain why SARS-CoV-2 spread so rapidly into and within nursing homes in Catalonia in a short period of time, as it also did in other European countries or the US [9, 28] resulting in substantial morbidity and mortality. These findings are strikingly similar to two studies of long-term care facilities where the outbreak was monitored from the very beginning [29, 30]. The findings suggest that asymptomatic transmission from SARS-CoV-2 residents most likely contributed to the rapid and extensive spread of infection to other residents and caregivers.

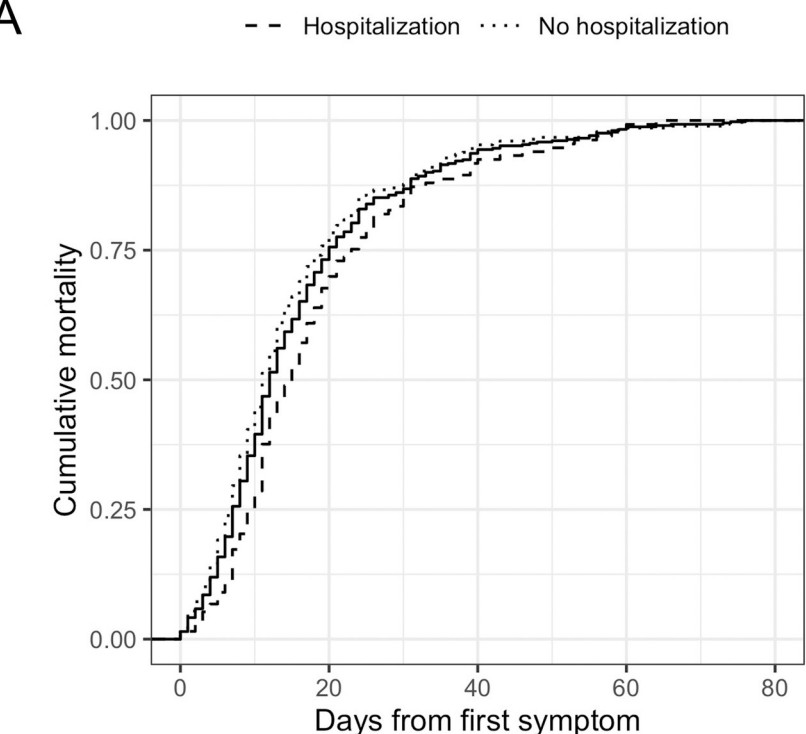

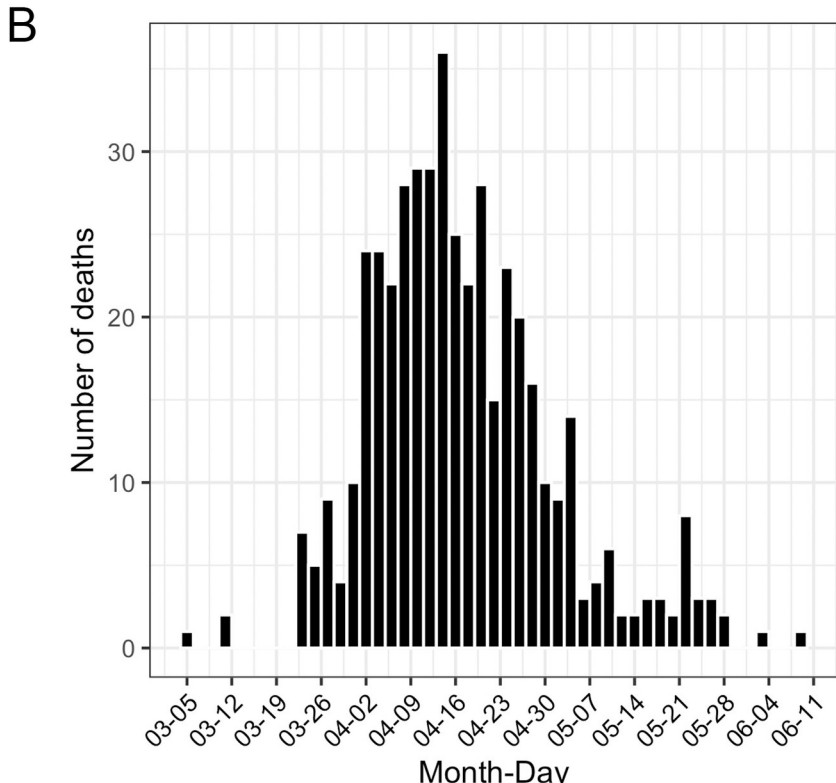

**Fig 2. Mortality in the cohort.** Legend: Panel A- Cumulative mortality rate. The p-value corresponds to the Kolmogorov-Smirnov test between the empirical cumulative distributions of survival days of hospitalized and non-hospitalized patients. Panel B- Date of exitus.

**Table 2. Risk factors associated with exitus.**

| Variable | Univariate OR (95CI) | Multivariate OR (95CI) |
|---|---|---|
| Age | 1.03 (1.02,1.05) | 1.03 (1.01, 1.05) |
| Pneumonia | 2.91 (2.26,3.73) | 1.74 (1.26, 2.38) |
| Fever | 4.04 (3.19,5.11) | 2.39 (1.80, 3.18) |
| Dyspnea | 5.61 (4.49,7.01) | 3.13 (2.37, 4.15) |
| Stupor | 26.32 (15.74,44) | 11.38 (6.54, 20.74) |
| Refusal to oral intake | 5.68 (4.4,7.33) | 3.21 (2.29, 4.51) |
| Diarrhea | 0.88 (0.66,1.18) | 0.61 (0.42, 0.88) |
| Mucous secretion | 2.76 (1.67,4.55) | 2.09 (1.05, 4.03) |
| Dry cough | 1.04 (0.8,1.35) | 0.74 (0.53, 1.04) |
| Eczema | 0.39 (0.14,1.11) | 0.36 (0.09, 1.02) |
| Asthenia | 2.69 (2.12,3.41) | 0.89 (0.63, 1.24) |
| Muscle pains | 1.6 (1.14,2.24) | 1.06 (0.66, 1.70) |
| Nasal Congestion | 1.54 (0.91,2.58) | 0.83 (0.38, 1.68) |
| Sore throat | 1.1 (0.56,2.18) | 1.14 (0.47, 2.58) |
| Vomiting | 1.26 (0.89,1.79) | 1.01 (0.64, 1.55) |
| Lip blisters | 7.22 (0.65,79.84) | 4.37 (0.27, 126.90) |
| Confusion | 1.89 (1.34,2.66) | 0.78 (0.49, 1.22) |
| Insomnia | 1.71 (0.73,3.98) | 0.86 (0.28, 2.43) |
| Dementia | 1.98 (1.58,2.49) | 1.63 (1.24, 2.16) |
| Hepatopathy | 0.6 (0.33,1.08) | 0.45 (0.20, 0.93) |
| Cardiovascular disease | 0.84 (0.69,1.04) | 0.81 (0.62, 1.05) |
| Cerebrovascular disease | 1.2 (0.95,1.53) | 1.10 (0.81, 1.48) |
| DM. without organic involvement | 1.17 (0.91,1.52) | 1.16 (0.84, 1.60) |
| DM. with organic involvement | 0.94 (0.64,1.39) | 0.96 (0.57, 1.57) |
| COPD | 1.19 (0.88,1.6) | 1.06 (0.735, 1.52) |
| Peptic ulcer | 1.6 (1.01,2.55) | 1.52 (0.85, 2.63) |
| CKD | 1.08 (0.85,1.37) | 0.89 (0.65, 1.20) |
| Connective tissue disease | 0.95 (0.6,1.5) | 0.74 (0.40, 1.33) |
| Cancer without metastases | 0.77 (0.54,1.1) | 0.93 (0.60, 1.40) |
| Cancer with metastases | 1.44 (0.45,4.62) | 0.85 (0.16, 3.50) |
| Hematological tumor | 0.24 (0.03,1.81) | 0.58 (0.03, 3.06) |

Legend: OR = odds ratio; CI = Confidence interval.

DM = Diabetes Mellitus; COPD = Chronic Obstructive Pulmonary Disease; CKD = Chronic Kidney Disease.

The proportion of asymptomatic infections reported in different studies varies greatly, ranging from 4% to 41% [31] and these differences may be explained among others by the definition of asymptomatic and paucisymptomatic or the presence of presymptomatic cases. A large nationwide, population-based seroepidemiological study in Spain of 61,000 individuals showed an average seroprevalence (study conducted from April 27th to May 11, 2020) of the infection of 5% in the region of Catalonia, which was slightly higher for nursing home workers (7.9%) [32]. This study also assessed the prevalence of COVID-19 related symptoms and concluded that asymptomatic cases represent between 22% and 36% of all SARS-CoV-2 infections in the general population. Our results showed a similar percentage of asymptomatic individuals (29%), although an earlier study evaluating 69 nursing homes in the metropolitan area of Barcelona suggested that 70% of the positive residents were

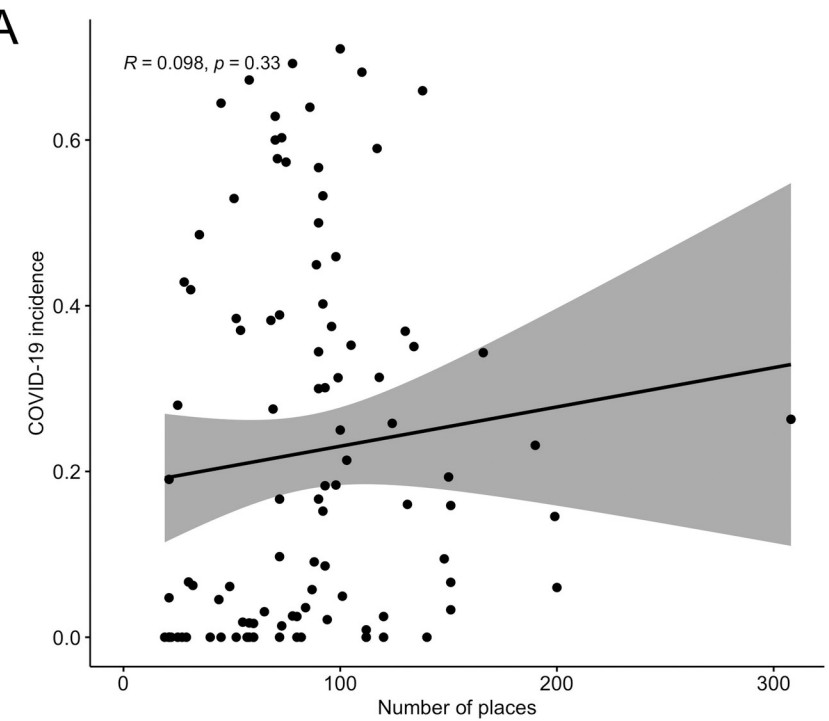

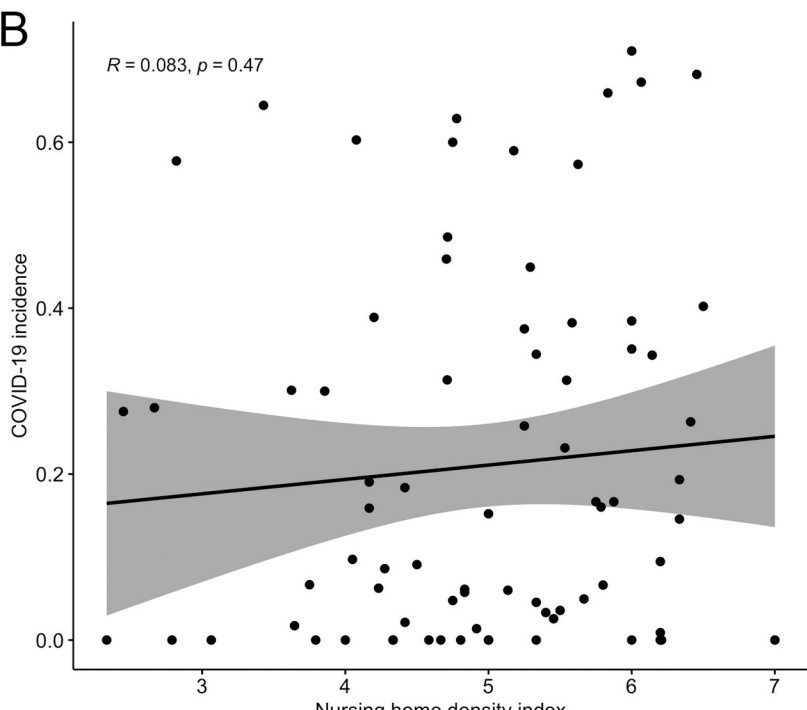

**Fig 3. Incidence vs density of the institution.** Legend: Panel A–Correlation between incidence of Covid 19 and number of places of the facility. Panel B–Correlation between incidence of Covid 19 and the nursing home density index.

asymptomatic [33]. This difference could be explained by the way results were reported, as that study did not record the onset of symptomatology after the PCR was conducted.

In a retrospective cross-sectional study of 14238 older people (>65 years) with confirmed COVID-19 in Wuhan during the first outbreak of the pandemic, 31% were diagnosed as severe or critical cases and only 1,4% were asymptomatic [34]. This contrasts with our results and the findings of a study performed in nursing homes in Connecticut where 28% residents tested positive and 22% were symptomatic or paucisymptomatic [35]. In the Wuhan case, the study was conducted during the first months of the pandemic, so most likely only severely affected cases were detected. Our results are consistent with those found in the Connecticut study where the setting (nursing homes) and the diagnostic strategy (active search of cases) were similar.

The range of symptoms in this study show a heterogeneous presentation: fever, dyspnea, cough, asthenia and diarrhea dominate the clinical presentation, and although the pairing fever-dyspnea was frequent. This heterogeneity must be considered when early alert protocols in nursing homes are developed as advanced age and the presence of comorbidities are associated with increased risk of mortality, and asymptomatic or paucisymptomatic patients have the potential for substantial viral shedding [36].

This retrospective cohort identified several risk factors for death in nursing home residents, where age shows a clear predictive value for mortality, an observation that is consistent with previous reports [37], and the typical presentation of fever and dyspnea also show a high predictive value in our model.

Results show that among all comorbidities, dementia had a high predictive value for death, defining a clinical presentation where fever, dyspnea, delirium and refuse to oral intake in a patient suffering from dementia represent a condition with very low probability for survival. Interestingly other comorbidities such as cardiovascular conditions and COPD, which are predictors of mortality in other studies [38] do not seem to have significant weight for poor prognosis. Geriatric syndromes complicate care and indicate a poor prognosis, so systematic assessment is imperative in ensuring adequate management and planning [39, 40]. In all cases, we found geriatric syndromes to be associated with a poorer prognosis in a similar manner to patients admitted in intermediate care facilities [41]. This association was most evident for dementia, delirium and inanition (refusal to oral intake with increased risk of malnutrition) showing that the presence of geriatric syndromes influenced the clinical evolution of patients with SARS-CoV-2 infection.

In a recent study of 351 nursing homes in the US analyzing 5256 residents with confirmed COVID-19, increased age, male gender, fever, shortness of breath, tachycardia, hypoxia, diabetes, chronic kidney disease, and impaired cognitive and physical function, were independently associated with mortality. Our findings also stressed the importance of age, some clinical symptoms, and dementia as risk factors associated with mortality [42].

De Vito et al. recently published a study describing the characteristics of the SARS-CoV-2 infection in an Italian cohort of patients living in nursing homes [43]. The demographic characteristics of their series was similar to our study and both series showed the same mortality rate in infected patients (21.2% in De Vito´s paper versus 21.75% in this study) regardless of the presence or absence of symptoms. Also the analysis of the clinical presentation in both series was similar with fever and dyspnea being the most common symptoms. Both studies highlight the importance of neurological involvement in the risk of developing COVID-19 and they suggest that the vulnerability of patients with dementia may be associated with a more critical presentation of COVID-19 [43]. However, the comparability of the risk factors between both series should be interpreted cautiously as the sample size was not comparable (264 versus 2092) including possible differences related to the demographic and social setting characteristics of both series that may account for different presentation of the disease.

Interestingly, we did not observe a significant effect on viral spreading in relationship to the size of the facility. Our observations suggest that, once the virus was introduced, rapid and wide-spread transmission occurred and small facilities had a similar incidence of infection to large ones. However, this observation must be interpreted cautiously as the limitation in the number of PCR tests did not rule out the possibility of the presence of additional asymptomatic carriers. Further studies are needed to better understand a potential relationship between widespread transmission and facility size as one of the strategies to impair the spread of the infection could be the implementation of small living units that act as "COVID-19 bubbles". In this scenario, a potential explanation could be that an efficient shielding effect may have been counteracted by the fact that the workforce performed transversally, providing care to all residents regardless of the living unit they were assigned to, thereby acting as spreaders within the facility [44].

## Conclusions

The high mortality rate in nursing homes highlights the vulnerability of this population and cumulative data demonstrates that asymptomatic and paucisymptomatic cases strongly contribute to the dynamics of transmission [45]. We have learned from past experience and also from the continuous outbreaks of COVID-19 in nursing homes that this is a foreseeable consequence of this pandemic. Therefore, we must implement strategies aimed at preventing the introduction of the pathogen into these facilities. To that end, it is imperative that public health authorities provide strategic guidance [46] and working protocols based on the monitoring of infection indicators and performing PCR screenings among residents and staff caring for them. These interventions must be accompanied by training staff in infection control, the use of PPEs and recognition of COVID-19 symptoms. A better understanding of COVID-19 in older adults living in nursing homes is necessary as many national and regional vaccination roll out strategies have put these facilities in the priority groups for vaccine administration and follow-up studies in this population are necessary to assess the efficiency of vaccination plans already in place. The potential emergence of future new variants of SARS-CoV-2 and the length of immune protection in this group of the population requieres a close monitoring of pandemic prepardness and response plans for nursing homes to protect this highly vulnerable population from this and other pathogens that may appear in the future.

## Limitations of the study

These limitations include (i) a potential bias in the profile of the nursing homes as their participation was established on a voluntary basis, (ii) the limited extend of the molecular diagnosis, in particular considering that, like in many other European regions, during the first outbreak of the pandemic there was a shortage of PCR tests and (iii) the lack of information on the treatment protocols due to the operational limitations of the data collection, a process directly related to the limited interoperability of the data systems between several health care providers (hospitals and primary care) involved in the implementation of treatment protocols once a patient was diagnosed by PCR in a particular nursing home. However, despite the limitations of the study, the mean incidence of the disease in this cohort does not significantly differ from the estimated global incidence in such facilities in Spain, suggesting that it represents a valid dataset for providing relevant information on the clinical characteristics of the disease in residents of nursing homes.

## Supporting information

**S1 File.**
(XLSX)

## Acknowledgments

We acknowledge the participating nursing homes as well as the residents and their families.

## Author Contributions

**Conceptualization:** Uxío Meis-Pinheiro, Francesc Lopez-Segui, Sandra Walsh, Anton Ussi, Sebastia Santaeugenia, Jose Augusto Garcia-Navarro, Antonio San-Jose, Antoni L. Andreu, Magda Campins, Benito Almirante.

**Data curation:** Uxío Meis-Pinheiro, Francesc Lopez-Segui, Antoni L. Andreu.

**Formal analysis:** Uxío Meis-Pinheiro, Francesc Lopez-Segui, Sandra Walsh, Anton Ussi, Sebastia Santaeugenia, Jose Augusto Garcia-Navarro, Benito Almirante.

**Investigation:** Uxío Meis-Pinheiro, Anton Ussi, Antonio San-Jose, Antoni L. Andreu, Magda Campins, Benito Almirante.

**Methodology:** Uxío Meis-Pinheiro, Sandra Walsh, Anton Ussi, Sebastia Santaeugenia, Antonio San-Jose, Antoni L. Andreu, Magda Campins, Benito Almirante.

**Supervision:** Sebastia Santaeugenia, Jose Augusto Garcia-Navarro, Antonio San-Jose, Antoni L. Andreu, Benito Almirante.

**Validation:** Sandra Walsh.

**Writing – original draft:** Uxío Meis-Pinheiro, Anton Ussi, Antoni L. Andreu, Benito Almirante.

**Writing – review & editing:** Uxío Meis-Pinheiro, Francesc Lopez-Segui, Sandra Walsh, Anton Ussi, Sebastia Santaeugenia, Jose Augusto Garcia-Navarro, Antonio San-Jose, Magda Campins, Benito Almirante.

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
