## [Decision Letter · Decision Letter 0]

21 Apr 2021

PONE-D-21-06498

CLINICAL CHARACTERISTICS OF COVID-19 IN OLDER ADULTS. A RETROSPECTIVE STUDY IN LONG-TERM NURSING HOMES IN CATALONIA

PLOS ONE

Dear Dr. ALMIRANTE,

Thank you for submitting your manuscript to PLOS ONE. After careful consideration, we feel that it has merit but does not fully meet PLOS ONE’s publication criteria as it currently stands. Therefore, we invite you to submit a revised version of the manuscript that addresses the points raised during the review process.

We look forward to receiving your revised manuscript.

Kind regards,

Giordano Madeddu

Academic Editor

PLOS ONE

Journal Requirements:

4. Please amend the manuscript submission data (via Edit Submission) to include author Benito Almirante. We note that he is twice on the list.

Reviewers' comments:

Reviewer's Responses to Questions

**Comments to the Author**

1. Is the manuscript technically sound, and do the data support the conclusions?

Reviewer #1: Partly

Reviewer #2: Yes

2. Has the statistical analysis been performed appropriately and rigorously? 

Reviewer #1: Yes

Reviewer #2: Yes

3. Have the authors made all data underlying the findings in their manuscript fully available?

Reviewer #1: No

Reviewer #2: Yes

4. Is the manuscript presented in an intelligible fashion and written in standard English?

Reviewer #1: No

Reviewer #2: No

5. Review Comments to the Author

Reviewer #1: Meis-Pinheiro et al. conducted an interesting retrospective study about clinical characteristics of COVID-19 in people living in long-term nursing homes. Many issues are present.

General comment

Abbreviations should be written entirely in the first appearance in the text (e.g. yr, CC).

English must be improved.

When the mean is reported, the authors should also write the standard deviation.

Introduction

The authors reported the number of infections and deaths in Spain in August. I suggest doing an update of these data.

The introduction is too short (15 lines). I suggest describing the SARS-CoV-2 and COVID-19, describing the clinical presentation of this disease, adding the major (fever, cough, dyspnea) and minor (anosmia, dysgeusia, headache, gastrointestinal symptoms, skin lesions). You could read and use these articles to improve the introduction: https://doi.org/10.26355/eurrev_202007_22291
https://doi.org/10.1002/hed.26269, https://doi.org/10.1002/hed.26204, https://doi.org/10.1016/S1473-3099(20)30402-3, https://doi.org/10.1097/IPC.0000000000000952 , https://doi.org/10.1111/eci.13427)

Furthermore, I suggest adding more information about the transmission of the disease, explaining why the nursing home is a risky setting and which factors could increase or decrease infection risk. I suggest reading and adding these manuscripts to your introduction: https://doi.org/10.1186/s40779-020-00240-0, https://doi.org/10.1001/jama.2020.12839, https://doi.org/10.26355/eurrev_202101_24424, https:/doi.org/10.1016/j.envpol.2020.115099, https://doi.org/10.1016/j.envpol.2020.11509.

Methods

In methods, the authors wrote "[…]provided retrospective data over a period of three months, from March 1st to May 31st, the period of the most severe impact of the pandemic wave in Spain.". Looking at the spanish data, the pandemic's most severe impact was in October-November 2020 and in January-February 2021. I suggest removing this sentence.

The patient population is not well defined. The entity referred to in the manuscript is retirement nursing homes. It is unclear if this refers to people in sheltered care/ warden controlled accommodation who require very little support, or residential Care home residents (requiring support with some daily living activities) or nursing home residents (requiring nursing care specifically, ie. a high degree of dependency).

How were symptoms ascertained? Were patients who had no symptoms reassessed to see if they were truly asymptomatic, vs pre-symptomatic?

Lines 90-99: this part is not clear. What do the authors mean with "spaces" and "unit"? what do the authors mean with "A", "B", "C", "D"?.

Why have the authors not considered anosmia and dysgeusia between the symptoms?

Result

I suggest not starting with "Table 1 shows demographic data, clinical symptoms of RT-PCR positive residents". It would be better to start with the cohort's description and use this sentence ad the end of the paragraph.

Lines 126-130: I suggest moving these lines in the method section.

I suggest replacing "average" with "mean". Furthermore, the standard deviation of the years is missing.

The sentence "However, the percentage was even higher when we only considered symptomatic individuals with a positive PCR (71% and 40%)", is not clear. Do not all people included in this study had a positive PCR for SARS-COV-2? (lines 78-80).

Lines 156-162. I suggest and the 95%CI after the OR.

It is not clear why the authors described the indicators to describe the facilities' structural characteristics if they have not used them in the results.

Furthermore, the number of deaths could have an impact on the SARS-CoV-2 infection and not on COVID-19.

No data about treatment are present in the manuscript. I suggest adding this information. Otherwise, you should explain the lack of information in the limitation.

Discussion

A recent study about Italian people living in retirement nursing homes has been published: https://doi.org/10.1371/journal.pone.0248009. I suggest reading it and using it to compare your data because there are some common points. In my opinion, this could increase the value of the discussion.

In the multivariate analysis, people with hepatopathy resulted having a lower mortality risk. In my opinion, this result should be discussed.

Table 1

About the p-value, I suggest using four after the comma, making an approximation in those p-values with 5 or 6 numbers after the comma.

It is not clear what "Excessive nasal discharge" means.

The authors sometimes used comma, sometimes dots, to divide the decimal numbers. I suggest always using dots.

I suggest replacing "Tumor without metastases" with "Solid tumor without metastases".

Regarding "AIDS", are you sure that all five people had AIDS and not HIV infection?

Table 2

I suggest switching the column Univariate and Multivariate.

Some words have been abbreviated without any reason (e.g. dis., inv., met.).

Figure 1 b. I suggest specifying that "fever", "dyspnea" means that those people had only that symptoms.

Reviewer #2: Meis-Pinheiro et al. aimed to describe clinical characteristics of COVID-19 in people living in long-term nursing homes. The topic is very interesting, given the subpopulation. However, there are numerous issues to point out:

Introduction

The authors reported the number of infections and deaths in Spain during August. Data are quite old and should be updated. Furthermore, the introduction is quite poor. Describing ways of transmission and clinical features is needed. Follow the example below for the structure and references (please, pay attention to the order):

1. Generalities

In December 2019, a new severe respiratory syndrome was identified in Wuhan, China. On January 2020, a new Coronavirus was detected and called SARS-CoV-2. On March 2021, the World Health Organization (WHO) declared SARS-CoV-2 disease (COVID-19) as a public health emergency.

2. Pathophysiology and transmission. https://doi.org/10.1186/s40779-020-00240-0;
https://doi.org/10.1001/jama.2020.12839;
https://doi.org/10.26355/eurrev_202101_24424;

3. Clinical features

Most common clinical features are fever, cough, dyspnea, and may also include anosmia, dysgeusia, headache, gastrointestinal symptoms, and skin lesions. https://doi.org/10.26355/eurrev_202007_22291;
https://doi.org/10.1002/hed.26269;
https://doi.org/10.1016/S1473-3099(20)30402-3

4. Why nursing homes must be evaluated? Explain the importance to provide an insight on this setting.

Methods

There are some not precise information in this section. For example, is reported ‘from March 1st to May 31st, the period of the most severe impact of the pandemic wave in Spain’. Please, delete this sentence. According to your national data, the most severe impact was in the last 5 months.

Readability is quite poor. Please, divide the Methods in subparagraph as follows:

1. Study population

This must be well defined. It is not clear the level of medical/nursing assistance needed in the setting (low, medium, high level of patient’s dependency). Are they sheltered care, residential care home residents, or nursing home residents?

2. Study conduction/assessment

Explain the kind of study (retrospective etc.) and your measures of evaluation. Furthermore, ‘spaces’, ‘units’, ‘A’, ’B’, ‘C’, ‘D’, ‘E’, ‘a’, ‘b’, ‘c’, ‘d’, ‘e’: there is low order in your methodology description. Please, better describe your variables.

3. Statistical analysis

Put before how you measured outcomes, and at the end the software.

4. Ethical issues

Put here your Ethical Committee authorization.

Among symptoms, anosmia and dysgeusia were not considered. Please, explain the reason. Furthermore, this could represent a limitation, with underestimation of symptomatic patients.

Result

Please, search to better present your results. First paragraph should be ‘Overall, xxx patients were included. Of them, xxx were female, and mean age was xxx + SD. Demographics, clinical features and health condition of included patients have been reported in table 1’.

P4L125-130: ‘The sample […] Catalonia’. This is your environment description and should be placed on Methods.

When reporting means (please, don’t use ‘average’), standard deviation must be added.

P4L131: you are starting your clinical features’ description referring to figure. Please, start with general description, then cite the figure.

There is something I still difficult to understand in the text. In Methods, there is a very long description of nursing homes’ structural characteristic. However, there is no mention or relationship studied regarding this parameters, clinical features, viral spread, or something else. This needs to be justified or the paragraph in Methods section should be deleted.

Is it possible to show data on patients’ treatment?

Discussion

A recent study with the same aim was published in PLOS ONE (https://doi.org/10.1371/journal.pone.0248009). Use this paper to enrich your discussion and eventually compare your results.

When looking the multivariate analysis, people with liver disease seemed to have lower mortality risk. Please, add a comment on this.

Tables and figures

Table 1

SD should be capitalized. When reporting p-value, remember to italicize. Furthermore, from the fourth number after the comma, you can approximate. Don’t use commas but points when reporting decimal numbers.

Instead of ‘Excessive nasal discharge’, please use ‘mucous secretion’ or something else. This is not a good description.

‘Tumor without metastases’ should be ‘cancer without metastases".

Table 2

Table 2 is on reverse. Please, report in the first column univariate and in the second multivariate analysis. Furthermore, delete the empty rows.

Figures

Please, when reporting figures delete ‘Title:’ (e.g. ‘Title: Clinical characteristics […]’ should be ‘Clinical features […]’)

Limitations section

Put a separate Limitations section after Conclusions. Beyond those still mentioned, some others should be mentioned:

- underestimation of symptomatic patients, given anosmia and dysgeusia were not considered

- lack of data regarding treatments (if you’ll not have possibility to show data). This is crucial. We don’t know how much people were treated, which were the drugs, and if there was a relationship with survival rates.

Language and typos

Please, carefully revise English language before resubmission. Abbreviations are not full written in the first appearance in the text (e.g. CC, yr, and so on). When reporting ‘p’ value, remember to italicize. Means are reported without standard deviation.

6. PLOS authors have the option to publish the peer review history of their article (what does this mean?). If published, this will include your full peer review and any attached files.

Reviewer #1: No

Reviewer #2: No

---

## [Author Response · Author response to Decision Letter 0]

28 May 2021

RESPONSE TO THE REVIEWERS

Reviewer #1: 

General comment

Abbreviations should be written entirely in the first appearance in the text (e.g. yr, CC).

This item has been fixed in the revised version 

English must be improved.

This new version has been edited by a native English-speaking coauthor of the manuscript.

When the mean is reported, the authors should also write the standard deviation.

In this update, SDs have been included when means are reported 

Introduction

The authors reported the number of infections and deaths in Spain in August. I suggest doing an update of these data.

Data of infections (590.000) and deaths (8.780) have been updated (April 2021) and the text has been edited accordingly 

The introduction is too short (15 lines). I suggest describing the SARS-CoV-2 and COVID-19, describing the clinical presentation of this disease, adding the major (fever, cough, dyspnea) and minor (anosmia, dysgeusia, headache, gastrointestinal symptoms, skin lesions). You could read and use these articles to improve the introduction: 

A new introduction includes a more detailed description of the clinical characteristics of COVID-19 including major and less frequent presentations. This description of the clinical characteristics of the disease has been referenced to the publications suggested by the reviewer.

Furthermore, I suggest adding more information about the transmission of the disease, explaining why the nursing home is a risky setting and which factors could increase or decrease infection risk. I suggest reading and adding these manuscripts to your introduction: 

A new paragraph describing the rationale of the increased risk of transmission in nursing homes has been added to the introduction and further discussed in the discussion section. In this part of the introduction the references suggested by the reviewer have been included and discussed to emphasize the importance of the setting that nursing homes represent for an increased risk of disease transmission.

Methods

In methods, the authors wrote "[…]provided retrospective data over a period of three months, from March 1st to May 31st, the period of the most severe impact of the pandemic wave in Spain.". Looking at the spanish data, the pandemic's most severe impact was in October-November 2020 and in January-February 2021. I suggest removing this sentence.

We agree that this expression is misleading and creates confusion for the reader. The sentence has now been changed to be precise to “the period of the first pandemic wave” 

The patient population is not well defined. The entity referred to in the manuscript is retirement nursing homes. It is unclear if this refers to people in sheltered care/ warden controlled accommodation who require very little support, or residential Care home residents (requiring support with some daily living activities) or nursing home residents (requiring nursing care specifically, ie. a high degree of dependency).

Again, we agree with the reviewer that the description of the population studied was poorly defined in the original version, in particular the degree of dependency. In this revised version we have clarified that the facilities analyzed (and, therefore, the population studied) were long-term nursing homes whose main objective is to provide integrate and full care to older adults with a high or very high degree of dependency.

How were symptoms ascertained? Were patients who had no symptoms reassessed to see if they were truly asymptomatic, vs pre-symptomatic?

We agree with the reviewer that we did not clarify which sources of data were used for the study and how symptomatology was assessed and reassessed. The revised version describes now that symptoms were annotated in individual Clinical History Forms and there was a daily reassessment to follow-up the presence or absence of a particular symptom or clinical sign.

Lines 90-99: this part is not clear. What do the authors mean with "spaces" and "unit"? what do the authors mean with "A", "B", "C", "D"?.

Indeed, the concept “space” and “units” were unclear. Instead of arbitrary “large” or “small” facilities and, with the aim of differentiating facilities where all residents shared the same spaces from those where daily life is organized in subgroups of residents sharing the same common spaces we created a synthetic indicator based on the number of “spaces” and “units”. “Spaces” were defined as rooms (excluding bedrooms), where residents spend time during the day (sitting areas, TV rooms, activity rooms, dining areas and so on). “Units” were defined as living units 4, the interconnected group of spaces where independent subgroups of residents do all their daily activities including bedrooms and common spaces used by a particular subgroup.

This is important as this concept is closely related to the level of social interactions within one specific facility. As an example, in a 100 place facility with no living units, all residents share the same common areas. In the same example of one facility of 100 places organized, but where daily life is organized, for example into 5 “units”, there will be 5 individual groups of 20 residents and interactions will tend to occur within these “bubbles” as the use of common spaces will be restricted to each particular bubble.

The concept has been now described into a format that it is easier to understand for the reader.

Why have the authors not considered anosmia and dysgeusia between the symptoms?

We agree with the reviewer that this needs to be clarified and a specific explanation has now been introduced in the novel version. The participants were old adults with a high or very high level of dependency, in most cases, mild to severe cognitive disorders, dementia and/or multiple underlying health conditions. This reality limited the value of subjectively perceived symptoms so to eliminate the possibility of bias we decided not to assess anosmia and dysgeusia in the study. 

Results

I suggest not starting with "Table 1 shows demographic data, clinical symptoms of RT-PCR positive residents". It would be better to start with the cohort's description and use this sentence ad the end of the paragraph.

We have now described the characteristics of the cohort and moved the sentence to the end of the paragraph. 

Lines 126-130: I suggest moving these lines in the method section.

Number of participant centers in cities with different population sizes has now been moved to the methods section

I suggest replacing "average" with "mean". Furthermore, the standard deviation of the years is missing.

This has now been changed and SD have been added

The sentence "However, the percentage was even higher when we only considered symptomatic individuals with a positive PCR (71% and 40%)", is not clear. Do not all people included in this study had a positive PCR for SARS-COV-2? (lines 78-80).

We agree with the reviewer that the sentence was misleading. Indeed all patients had a positive PCR but not all of them showed symptoms so we think it is relevant to mention the frequency of fever and dyspnea in the whole series of positive PCR individuals, highlighting that when only symptomatic patients were considered, the frequency of these two symptoms increases. We have now reformulated the expression to make it clearer to the reader

Lines 156-162. I suggest and the 95%CI after the OR.

The change has been done

It is not clear why the authors described the indicators to describe the facilities' structural characteristics if they have not used them in the results.

We have clarified this issue in the new version, specifying “places” instead of “beds” referring to Fig 3a and referring to the synthetic indicator that now is precisely described in the amended Methods section.

No data about treatment are present in the manuscript. I suggest adding this information. Otherwise, you should explain the lack of information in the limitation.

We agree that this is a relevant aspect but collecting data on treatment was beyond the scope of this study due to the complex interoperability in the data systems that came about because of the high number of health care providers as patients were followed-up by many different medical teams and hospitals of the region. However, we agree with the reviewer that this consideration is relevant for the reader, so we have restructured the paragraph of limitations of the study in the discussion section to highlight this observation.

Discussion

A recent study about Italian people living in retirement nursing homes has been published: https://doi.org/10.1371/journal.pone.0248009. I suggest reading it and using it to compare your data because there are some common points. In my opinion, this could increase the value of the discussion.

We agree with the reviewer that the study published by De Vito et al. is highly relevant for the discussion of our results and the strategy approach of both studies are similar. Although De Vito´s study was able to collect data on medical treatments and ours was not for the reasons explained above, it is interesting to mention that both series converge of the similar conclusion that the high transmission rate of SARS-CoV2 infection in nursing homes could be related to crowding, sharing of gathering areas, and inadequate infection prevention and control measures. Although the different size of the cohorts (264 patients in De Vito´s paper and 2092 in our study) could explain some differences in the multivariate analysis of the factors that influenced infection occurrence, it is clear that there are strong similarities between both studies such as the mortality rate or the presence of neurological syndromes as a risk factor for developing COVID-19 symptomatic disease. Following the suggestion of the reviewer De Vito´s paper has now been quoted and the similarities between both studies has been discussed in the discussion section.

In the multivariate analysis, people with hepatopathy resulted having a lower mortality risk. In my opinion, this result should be discussed.

Indeed this a striking result considering that several studies have shown that COVID-19 patients with preexisting liver diseases face a higher risk of decompensation and mortality (Mohammed A, Paranji N). However most of these studies have been carried out in adult patients and little data is available on the response of liver function in older adults affected not only by SARS-CoV-2 infection but also other viral pathogens. We have included in this discussion the observation of Kondo et al. (Kondo Y, Tsukada K, Hepatology) who described during an outbreak of acute HBV in nursing home residents, most infected patients were asymptomatic and no patients died or required hospitalization suggesting that more studies 

are needed to understand the role of liver function in the elderly in response to viral pathogens.

Table 1

About the p-value, I suggest using four after the comma, making an approximation in those p-values with 5 or 6 numbers after the comma.

It is not clear what "Excessive nasal discharge" means.

The authors sometimes used comma, sometimes dots, to divide the decimal numbers. I suggest always using dots.

I suggest replacing "Tumor without metastases" with "Solid tumor without metastases".

Regarding "AIDS", are you sure that all five people had AIDS and not HIV infection?

The indications suggested by the reviewer have been incorporated. The term “excessive nasal discharge” has been substituted by “rhinorrhea” and for AIDS, we have changed to “HIV infection” as we did not collect precise information of their immune status or the protocol for antiretroviral therapy. 

Table 2

I suggest switching the column Univariate and Multivariate.

Some words have been abbreviated without any reason (e.g. dis., inv., met.).

Changes suggested in Table 2 have now been introduced in this version 

Figure 1 b. I suggest specifying that "fever", "dyspnea" means that those people had only that symptoms.

This has been clarified in the new version

Reviewer #2:

Meis-Pinheiro et al. aimed to describe clinical characteristics of COVID-19 in people living in long-term nursing homes. The topic is very interesting, given the subpopulation. However, there are numerous issues to point out:

Introduction

The authors reported the number of infections and deaths in Spain during August. Data are quite old and should be updated. Furthermore, the introduction is quite poor. Describing ways of transmission and clinical features is needed. Follow the example below for the structure and references (please, pay attention to the order):

1. Generalities

In December 2019, a new severe respiratory syndrome was identified in Wuhan, China. On January 2020, a new Coronavirus was detected and called SARS-CoV-2. On March 2021, the World Health Organization (WHO) declared SARS-CoV-2 disease (COVID-19) as a public health emergency.

2. Pathophysiology and transmission. https://doi.org/10.1186/s40779-020-00240-0;
https://doi.org/10.1001/jama.2020.12839;
https://doi.org/10.26355/eurrev_202101_24424;

3. Clinical features

Most common clinical features are fever, cough, dyspnea, and may also include anosmia, dysgeusia, headache, gastrointestinal symptoms, and skin lesions. https://doi.org/10.26355/eurrev_202007_22291;
https://doi.org/10.1002/hed.26269;
https://doi.org/10.1016/S1473-3099(20)30402-3

4. Why nursing homes must be evaluated? Explain the importance to provide an insight on this setting.

We appreciate the suggestion of restructuring the introduction to present in a coherent way the characteristics of COVID-19 in the particular context of the elderly and, specifically, in the nursing home setting. Following the indications of the reviewer we have created a new introduction based on a narrative that describes the general description of SARS-CoV-2 infection, the clinical characteristics and the particularities of the nursing homes setting for an increased risk of transmission We have included some of the references suggested by the reviewer that will help the reader to contextualize our study.

Methods

There are some not precise information in this section. For example, is reported ‘from March 1st to May 31st, the period of the most severe impact of the pandemic wave in Spain’. Please, delete this sentence. According to your national data, the most severe impact was in the last 5 months.

We have updated the data to April 2021 and corrected the expression “most severe pandemic wave” for “first pandemic wave”. Although in terms of mortality this first wave was by far the most deadly one, we agree that the concept “severity” in the context of our description is confusing and it is more sound to refer to the timeline of the pandemic outbreak.

Readability is quite poor. Please, divide the Methods in subparagraph as follows:

1. Study population

This must be well defined. It is not clear the level of medical/nursing assistance needed in the setting (low, medium, high level of patient’s dependency). Are they sheltered care, residential care home residents, or nursing home residents?

2. Study conduction/assessment

Explain the kind of study (retrospective etc.) and your measures of evaluation. Furthermore, ‘spaces’, ‘units’, ‘A’, ’B’, ‘C’, ‘D’, ‘E’, ‘a’, ‘b’, ‘c’, ‘d’, ‘e’: there is low order in your methodology description. Please, better describe your variables.

3. Statistical analysis

Put before how you measured outcomes, and at the end the software.

4. Ethical issues

Put here your Ethical Committee authorization.

Among symptoms, anosmia and dysgeusia were not considered. Please, explain the reason. Furthermore, this could represent a limitation, with underestimation of symptomatic patients.

We acknowledge the suggestion of the reviewer for structuring the methods section into different subsections. The new version has included this new structure.

Indeed, the concept “space” and “units” were unclear. Instead of arbitrary “large” or “small” facilities and, with the aim of differentiating facilities where all residents shared the same spaces from those where daily life is organized in subgroups of residents sharing the same common spaces we created a synthetic indicator based on the number of “spaces” and “units”. “Spaces” were defined as rooms (excluding bedrooms) where residents spend time during the day (sitting areas, TV rooms, activity rooms, dining areas and so on). “Units” were defined as living units 4, the interconnected group of spaces where independent subgroups of residents do all their daily activities, including bedrooms and common spaces used by a particular subgroup.

This is important as this concept is closely related to the level of social interactions within one specific facility. As an example, in a 100-place facility with no living units, all residents share the same common areas. In the same example of one facility of 100 places where daily life is organized, for example on 5 “units”, there will be 5 individual groups of 20 residents and interactions will tend to occur within these “bubbles” as the use of common spaces will be restricted to each particular bubble.

On the comment of the lack of information on the anosmia/dysgeusia, we agree with the reviewer that this needs to be clarified and a specific explanation has now been introduced. The participants were old adults with a high or very high level of dependency, in most cases, mild to severe cognitive disorders, dementia and/or multiple underlying health conditions. This reality limited the value of subjectively perceived symptoms so to eliminate the possibility of bias we decided not to assess anosmia and dysgeusia in the study.

Results

Please, search to better present your results. First paragraph should be ‘Overall, xxx patients were included. Of them, xxx were female, and mean age was xxx + SD. Demographics, clinical features and health condition of included patients have been reported in table 1’.

P4L125-130: ‘The sample […] Catalonia’. This is your environment description and should be placed on Methods.

When reporting means (please, don’t use ‘average’), standard deviation must be added.

P4L131: you are starting your clinical features’ description referring to figure. Please, start with general description, then cite the figure.

There is something I still difficult to understand in the text. In Methods, there is a very long description of nursing homes’ structural characteristic. However, there is no mention or relationship studied regarding this parameters, clinical features, viral spread, or something else. This needs to be justified or the paragraph in Methods section should be deleted.

Is it possible to show data on patients’ treatment?

We have restructured the results sections clarifying the questions raised by the reviewers. These include:

- We start describing the characteristics of the cohort and the environment descriptions have been moved to the methods sections, including the number of participants in cities with different population sizes.

- “Average” has been replaced by “Mean” and SD have been added.

- The sentence "However, the percentage was even higher when we only considered symptomatic individuals with a positive PCR (71% and 40%)" was misleading. Indeed all patients had a positive PCR but not all of them showed symptoms so we think it is relevant to mention the frequency of fever and dyspnea in the whole series of positive PCR individuals, highlighting that when only symptomatic patients were considered, the frequency of these two symptoms increases. We have now reformulated the expression to make it clearer to the reader.

- We have added 95%CI after the OR.

- We have now explained in detail the bases for the analysis of the structural characteristics of the facilities. In this novel version we use the term “places” instead of “beds” referring to Fig 3a and referring to the synthetic indicator that now is precisely described in the amended Methods section.

- We appreciate the comment about the data on patient treatment. This is a relevant aspect but collecting data on treatment was beyond the scope of this study due to the complex interoperability in the data systems that came about because of the high number of health care providers as patients were followed-up by many different medical teams and hospitals of the region. However, we agree with the reviewer that this consideration is relevant for the reader, so we have restructured the paragraph of limitations of the study in the discussion section to highlight this observation.

Discussion

A recent study with the same aim was published in PLOS ONE (https://doi.org/10.1371/journal.pone.0248009). Use this paper to enrich your discussion and eventually compare your results.

We agree with the reviewer that the study published by De Vito et al. is highly relevant for the discussion of our results and the strategy approach of both studies are similar. Although De Vito´s study was able to collect data on medical treatments and ours was not for the reasons explained above, it is interesting to mention that both series converge of the similar conclusion that the high transmission rate of SARS-CoV2 infection in nursing homes could be related to crowding, sharing of gathering areas, and inadequate infection prevention and control measures. Although the different size of the cohorts (264 patients in De Vito´s paper and 2092 in our study) could explain some differences in the multivariate analysis of the factors that influenced infection occurrence, it is clear that there are strong similarities between both studies such as the mortality rate or the presence of neurological syndromes as a risk factor for developing COVID-19 symptomatic disease.

When looking the multivariate analysis, people with liver disease seemed to have lower mortality risk. Please, add a comment on this.

Indeed this a striking result considering that several studies have shown that COVID-19 patients with preexisting liver diseases face a higher risk of decompensation and mortality (Mohammed A, Paranji N). However most of these studies have been carried out in adult patients and little data is available on the response of liver function in older adults affected not only by SARS-CoV-2 infection but also other viral pathogens. We have included in this discussion the observation of Kondo et al. (Kondo Y, Tsukada K, Hepatology) who described during an outbreak of acute HBV in nursing home residents, that most infected patients were asymptomatic and no patients died or required hospitalization suggesting that more studies are needed to understand the role of liver function in the elderly in response to viral pathogens.

Tables and figures

Table 1

SD should be capitalized. When reporting p-value, remember to italicize. Furthermore, from the fourth number after the comma, you can approximate. Don’t use commas but points when reporting decimal numbers.

Instead of ‘Excessive nasal discharge’, please use ‘mucous secretion’ or something else. This is not a good description.

‘Tumor without metastases’ should be ‘cancer without metastases".

We have included in this revised version the suggestions of the reviewer. The expression “Excessive nasal discharge” has been replaced by “rhinorrhea”.

Table 2

Table 2 is on reverse. Please, report in the first column univariate and in the second multivariate analysis. Furthermore, delete the empty rows.

The structure of table 2 has been modified following the reviewer´s suggestion

Figures

Please, when reporting figures delete ‘Title:’ (e.g. ‘Title: Clinical characteristics […]’ should be ‘Clinical features […]’)

This change has now been introduced in the revised version

Limitations section

Put a separate Limitations section after Conclusions. Beyond those still mentioned, some others should be mentioned:

- underestimation of symptomatic patients, given anosmia and dysgeusia were not considered

- lack of data regarding treatments (if you’ll not have possibility to show data). This is crucial. We don’t know how much people were treated, which were the drugs, and if there was a relationship with survival rates.

Following the reviewer´s suggestion, we have added a section on “limitations on the study”, emphasizing that, in the context of the abrupt outbreak of COVID-19 in long-term nursing homes in Catalonia in April 2020 (interoperability issues, several health care providers involved) it was not possible to collect information of good quality on the treatments received by SARS-CoV-2 patients who develop COVID-19. Also we believe that in the current structure of the manuscript the limitation on the information of the presence of anosmia/dysgeusia fit better in the methods section so this limitation has been mentioned there.

Language and typos

Please, carefully revise English language before resubmission. Abbreviations are not full written in the first appearance in the text (e.g. CC, yr, and so on). When reporting ‘p’ value, remember to italicize. Means are reported without standard deviation.

English has been carefully revised and comments on abbreviations, “p” value and report of means and SD have now been fixed in this revised version

---

## [Decision Letter · Decision Letter 1]

29 Jun 2021

PONE-D-21-06498R1

CLINICAL CHARACTERISTICS OF COVID-19 IN OLDER ADULTS. A RETROSPECTIVE STUDY IN LONG-TERM NURSING HOMES IN CATALONIA

PLOS ONE

Dear Dr. Almirante,

Thank you for submitting your manuscript to PLOS ONE. After careful consideration, we feel that it has merit but does not fully meet PLOS ONE’s publication criteria as it currently stands. Therefore, we invite you to submit a revised version of the manuscript that addresses the points raised during the review process.

We look forward to receiving your revised manuscript.

Kind regards,

Giordano Madeddu

Academic Editor

PLOS ONE

Journal Requirements:

Reviewers' comments:

Reviewer's Responses to Questions

**Comments to the Author**

1. If the authors have adequately addressed your comments raised in a previous round of review and you feel that this manuscript is now acceptable for publication, you may indicate that here to bypass the “Comments to the Author” section, enter your conflict of interest statement in the “Confidential to Editor” section, and submit your "Accept" recommendation.

Reviewer #1: All comments have been addressed

Reviewer #2: All comments have been addressed

2. Is the manuscript technically sound, and do the data support the conclusions?

Reviewer #1: Yes

Reviewer #2: Yes

3. Has the statistical analysis been performed appropriately and rigorously? 

Reviewer #1: Yes

Reviewer #2: Yes

4. Have the authors made all data underlying the findings in their manuscript fully available?

Reviewer #1: No

Reviewer #2: (No Response)

5. Is the manuscript presented in an intelligible fashion and written in standard English?

Reviewer #1: Yes

Reviewer #2: Yes

6. Review Comments to the Author

Reviewer #1: The authors provide to assess the manuscript as suggested by my previous revision. Some minor issues are present:

Abbreviations should be written entirely in the first appearance in the text (e.g., COVID-19).

I suggest modifying the first part of the introduction. In my opinion, the authors should start writing a sentence about what is SARS-CoV-2 before explaining the symptoms caused by it.

The figures are missing in this new version of the manuscript. Furthermore, there is no references in the text for the figures. I suggest adding them.

Reviewer #2: The athours thoroughly revised their manuscript and I thank them for addressing my comments. The paper is now ready for publication on PLOS ONE.

7. PLOS authors have the option to publish the peer review history of their article (what does this mean?). If published, this will include your full peer review and any attached files.

Reviewer #1: No

Reviewer #2: No

---

## [Author Response · Author response to Decision Letter 1]

7 Jul 2021

RESPONSE TO THE REVIEWERS

6. Review Comments to the Author

Reviewer #1: The authors provide to assess the manuscript as suggested by my previous revision. Some minor issues are present:

Abbreviations should be written entirely in the first appearance in the text (e.g., COVID-19).

This item has been fixed in the revised version.

I suggest modifying the first part of the introduction. In my opinion, the authors should start writing a sentence about what is SARS-CoV-2 before explaining the symptoms caused by it.

In this update, this part has been addressed with a new introductory text.

The figures are missing in this new version of the manuscript. Furthermore, there is no references in the text for the figures. I suggest adding them.

This item has been fixed in the revised version and the figures added.

---

## [Editor Report · Decision Letter 2]

12 Jul 2021

CLINICAL CHARACTERISTICS OF COVID-19 IN OLDER ADULTS. A RETROSPECTIVE STUDY IN LONG-TERM NURSING HOMES IN CATALONIA

PONE-D-21-06498R2

Dear Dr. Almirante,

We’re pleased to inform you that your manuscript has been judged scientifically suitable for publication and will be formally accepted for publication once it meets all outstanding technical requirements.

Kind regards,

Giordano Madeddu

Academic Editor

PLOS ONE
---

## [Editor Report · Acceptance letter]

16 Jul 2021

PONE-D-21-06498R2 

CLINICAL CHARACTERISTICS OF COVID-19 IN OLDER ADULTS.
A RETROSPECTIVE STUDY IN LONG-TERM NURSING HOMES IN CATALONIA 

Dear Dr. Almirante:

I'm pleased to inform you that your manuscript has been deemed suitable for publication in PLOS ONE. Congratulations! Your manuscript is now with our production department. 

Kind regards, 

on behalf of

Dr. Giordano Madeddu 

Academic Editor

PLOS ONE